# Epstein-Barr Virus-Positive Inflammatory Follicular Dendritic Cell Sarcoma Presenting as a Colonic Polyp: Report of a Case with a Literature Review

**DOI:** 10.3390/medicina59071341

**Published:** 2023-07-21

**Authors:** Jiahui Hu, Dongdong Huang, Chengfu Xu, Yi Chen, Han Ma, Zhe Shen

**Affiliations:** 1Department of Gastroenterology, The First Affiliated Hospital, Zhejiang University School of Medicine, Hangzhou 310003, China; 22218368@zju.edu.cn (J.H.); xiaofu@zju.edu.cn (C.X.); zyyyychen@zju.edu.cn (Y.C.); shenzdr@zju.edu.cn (Z.S.); 2Department of Pathology, The First Affiliated Hospital, Zhejiang University School of Medicine, Hangzhou 310003, China; hdd0808@zju.edu.cn

**Keywords:** follicular dendritic cell sarcoma, inflammatory, colon polyp, Epstein-Barr virus, lgG4

## Abstract

*Background*: Follicular dendritic cell (FDC) sarcoma is an uncommon mesenchymal origin neoplasm derived from the abnormal proliferation and differentiation of FDCs. Epstein‒Barr virus-positive inflammatory follicular dendritic cell sarcoma (EBV+ iFDCS), which used to be known as the inflammatory pseudotumour (IPT)-like variant, occurs exclusively in the liver and spleen and has rarely been reported in the gastrointestinal tract. *Case study*: Here, we report a case of a 52-year-old woman with a special family history undergoing a routine physical examination. The colonoscope revealed an approximately 18 mm transverse colonic polyp, and the endoscopic polypectomy was performed. Microscopically, the excised polypoid mass was composed predominantly of inflammatory cells scattered with atypical ovoid to spindle tumor cells. Interestingly, there was a remarkable infiltration of IgG4+ cells. Immunohistochemistry showed that the tumor cells were positive for CD21, CD23 and CD35. EBV-encoded mRNA (EBER) in situ hybridization also gave positive signals. These histopathology features supported the diagnosis of EBV+ iFDCS. The patient was free of disease over 1-year follow-up. *Conclusion*: Identification of the potential pathogenesis sites of EBV+ iFDCS in extra-hepatosplenic regions is necessary for correct and timely diagnosis, and we consider it very meaningful to share our experience of diagnosing this tumor type. Furthermore, we summarize the clinicopathological features of EBV+ iFDCS presenting as a colon polyp after a thorough review of the literature.

## 1. Introduction

Follicular dendritic cells (FDCs) are mesenchymal-derived cells located in B follicles that serve as antigen-presenting cells and play a major role in the induction and maintenance of the humoral immune response [1]. Follicular dendritic cell sarcoma (FDCS) is a rare neoplasm that derives from the abnormal proliferation and differentiation of FDCs [2]. It was first described by Monda et al. [3] in 1986, reporting four cases with features suggesting a dendritic reticulum cell origin.

FDCs can be divided into the conventional type and inflammatory pseudotumour (IPT)-like variant type according to their histological morphology. However, Epstein-Barr virus (EBV)-positive inflammatory FDCS (EBV+ iFDCS), which used to be classified as the latter type, is delineated as an entity separate from FDCS given its remarkable clinicopathologic features according to the fifth edition of the World Health Organization (WHO) Classification of Hematolymphoid Tumors [4]. It is distinguished by its prominent inflammatory component and positivity for EBV by in situ hybridization [2,5]. Similar to FDCS, EBV+ iFDCS cells are immunoreactive for various specific markers, such as CD21, CD35, CD23 and CNA.42 [6].

The incidence of EBV+ iFDCS has marked female predominance (female to male ratio = 1.71:1) according to report [7]. And EBV+ iFDCS is reported mostly in the spleen and liver [6]. However, the clinicopathological landscape of this disease has been broadened by the discovery of cases with extrahepatosplenic involvement, such as in the gastrointestinal (GI) tract, palatine or nasopharyngeal tonsils and peripancreatic region [2,6,8]. To date, there have been only 12 cases of EBV+ iFDCS presenting as a colon polyp reported in the literature (Table 1). Here, we report a case of EBV+ iFDCS in a 52-year-old woman who presented with a transverse colon polyp with no obvious clinical symptoms.

## 2. Case Presentation

A 52-year-old woman was found to have a colonic polyp during a routine endoscopic examination two weeks before admission. No abdominal pain, bowel habit change, body weight loss or fever was noted. Physical examination revealed no evidence of lymphadenopathy or abdominal tenderness. There was no particular personal medical history. The fecal occult blood test was negative. Laboratory data including hemoglobin, tumor markers, renal function and liver function tests were all within normal limits. Gastroscopic examination suggested chronic atrophic gastritis. Colonoscopic examination in our hospital revealed an approximately 18 mm polyp 570 mm from the anal verge, with congestion of the mucosa overlying the polyp (Figure 1). According to the Paris classification of colorectal polyp, we classified the lesion as type 0-I. Endoscopic biopsy of the polyp prior to admission revealed inflammatory granulation tissue with focal lymphoid hyperplasia. Based on the endoscopic morphological characteristics of the lesion, the positive result of the lifting sign and previous biopsy results, endoscopic polypectomy was performed, and the resected tumor was approximately 1.2 × 0.9 × 0.6 cm.

Microscopically, the excised polypoid mass showed obvious lymphoid tissue and plasma cell infiltration in the mucosal and submucosal layers, scattered with atypical ovoid to spindle tumor cells, which demonstrated vesicular chromatin and clear nuclei but indistinct cell borders (Figure 2A–C). The nuclear atypia varied from mild to moderate, and some occasionally seen paired nuclei. Immunohistochemical analysis indicated that the tumor cells were variably immunoreactive for CD21 (Figure 2D), CD23 (Figure 2E) and CD35 (Figure 2F). The dense lymphoplasmacytic infiltration background was mixed with T cells (positive for CD3 and CD5) and B cells (positive for CD20), with a predominance of the B-lineage. CD10 and bcl-6 highlighted follicular germinal centers, while MUM-1 and kappa/lambda staining revealed polyclonal plasma cells. Notably, EBV-encoded mRNA (EBER) in situ hybridization (ISH) also showed positive signals on the various-sized ovoid to spindle tumor cells (Figure 2G).

Interestingly, there was a remarkable increase in IgG4+ cells (>100/HPF), accompanied by an elevated IgG4: IgG ratio (40~50%) (Figure 2H,I). However, typical pathological features of IgG4-related disease (IgG4-RD), such as obliterative phlebitis and storiform fibrosis, were lacking. Additionally, her postoperative serum IgG4 was normal (1.12 g/L, reference range 0.030–2.010). Unfortunately, her preoperative serum IgG4 was unavailable.

Subsequent abdominal contrast-enhanced computed tomography (CT) showed no obvious space-occupying lesions. Positron emission tomography/computed tomography (PET/CT) showed a metal clip shadow in the transverse colon operation area, without obvious thickening of the intestinal wall or any abnormally increased radioactive uptake. It also showed an approximately 8 × 7 mm slightly low-density nodule in the left thyroid with increased fluorodeoxyglucose (FDG) metabolism, and the maximum SUV was approximately 14.2 (Figure 3). However, the subsequent fine needle aspiration (FNA) of the thyroid nodule found well-differentiated follicular epithelial cells. She was free of disease over 1-year follow-up after the endoscopic polypectomy without further treatment.

Notably, we were surprised to learn that her son was diagnosed with a myeloid/lymphoid neoplasm with eosinophilia (MLN-EO) with FIP1L1-platelet-derived growth factor receptor alpha (PDGFRA) rearrangement manifesting as T lymphoblastic lymphoma (T-LBL) for over a year before proper diagnosis, a condition with only approximately 10 known cases worldwide.

## 3. Discussion

In brief, we report a case of a 52-year-old woman diagnosed with EBV+ iFDCS presenting as a transverse colonic polyp. Specifically, there was a remarkable infiltration of IgG4+ cells on histopathology. She also had an abnormality of the thyroid gland, and her family history of tumors is unique, making us more interested in exploring the cause of her tumor further.

As a rare tumor with several clinicopathologic characteristics, EBV+ iFDCS is always hard to diagnose especially when it presents as a gastrointestinal polyp. Patients are usually asymptomatic or only present with some non-specific symptoms including abdominal pain or hematochezia, which makes the identification more difficult for clinicians. Also, according to the endoscopic findings, its morphological characteristics are not very different from those of general hyperplastic polyps. And it can be effectively identified only when combined with pathology. We therefore hope that the diagnostic experience in our institution will allow more clinicians to recognize the specific phenotype of the disease, thus avoiding misdiagnosis or inappropriate treatment.

The clinical and pathological characteristics of EBV+ iFDCS presenting as a colon polyp, including the current case, are summarized in Table 1. There have been only 12 cases reported in the literature thus far [5,8,9,10,11,12]. The patients’ ages ranged from 46 to 78 years, and the mean and median were 58.8 and 56 years, respectively, without any evident sex predominance (female: male = 7:6). All cases were variably immunoreactive with a panel of FDC markers, including CD21, CD23, CD35 and D2-40. The colonic lesions almost pursued an indolent course following polypectomy or resection, except for one case of coexisting paraneoplastic pemphigus, which eventually became the cause of death [8]. Surprisingly, one case mentioned peculiar increased lgG4+ plasma cell (240/HPF) infiltration similar to ours [10], inspiring us to explore the potential role of lgG4+ plasma cells in EBV+ iFDCS.

In the current case, although the patient’s postoperative serum IgG4 was normal and there were no radiological findings related to IgG4-RD, the pathology revealed a remarkable increase in IgG4+ cells (>100/HPF), accompanied by an elevated IgG4: IgG ratio (40~50%). Studies have found that EBV reactivation can stimulate the plasma cell differentiation of host B cells and Ig production, leading to IgG4-positive plasma cell infiltration and high serum IgG4 levels [13]. Choe et al. [14] reported that IgG4+ plasma cells were notably increased in six cases of splenic EBV+ iFDCS. In that study, the numbers of IgG4+ cells in each tumor ranged from 27 to 128 per high-power field, and the ratio of IgG4+/IgG+ plasma cells was 25 to 75%. Histological manifestations showed obliterative vasculitis (3/6, 50%) and sclerosis (4/6, 66.7%). However, whether lgG4 mediates the genesis and development of EBV+ iFDCS needs further confirmation.

In consideration of the universal infection of EBV in this tumor, it is of vital importance to discover its pathogenetic role. EBV infects over 95% of the world’s population [15]. Research has shown that female adults are more likely to be seropositive than males in the prevalence of EBV-IgG-antibodies (98.6% vs. 95%, *p* < 0.01). And they found that EBV antibody titers are generally higher in females, while other studies have indicated that the seroprevalence of EBV is the same among both sexes at any age [16,17]. EBV is mainly transmitted through saliva. However, breast milk, bodily fluids and transplantation of EBV-positive organs can also spread the virus from one to another [15]. It has been confirmed that EBV-induced chemokines and monokines, such as interferon-gamma-inducible protein-10 (IP-10) and monokine induced by interferon-gamma (Mig) have been implicated in the induction of vascular damage and tissue necrosis [18]. Furthermore, EBV-encoded latent membrane protein 1 (EBV-LMP1), a potential carcinogenic protein, has been frequently identified in cases of EBV+ iFDCS. Currently, there is no specific cure or preventive drug for EBV. EBV prophylactic vaccines and therapeutic vaccines have been in development for many years. Although neither one has been licensed, there have been very encouraging results with several EBV vaccine prototypes [19]. However, a few cases of EBV-negative inflammatory FDCS have been reported in recent years, suggesting that other related pathogenesis exists and requires further investigation [20].

As EBV-associated disorders including EBV+ iFDCS tend to affect females, some researchers suggested that sex-dependent pathogenic responses to EBV may contribute to susceptibility or progression of these disorders. One study detected the gain of chromosome X by SNP array, which was considered clonal in a patient diagnosed with EBV+ iFDCS [21]. Meanwhile, the existing data from cancer genomes indicate that the female X chromosome, particularly inactive X, usually has a higher mutation burden of point mutations and that this is a frequent event in cancer [22]. However, as the molecular pathogenesis of EBV+ iFDCS is not clear, the specific reasons for the gender difference need to be further studied.

Several other tumors share similar histological morphology with EBV+ iFDCS, presenting challenges for doctors to distinguish between them. Inflammatory fibroid polyp (IFP) is a vital morphological mimic of EBV+ iFDCS, characterized by spindle stromal cells and intermixed inflammatory cell infiltration, predominantly eosinophils on histology [5]. Both spindle cells and eosinophils may concentrate around the distinctive rich vascular network. The spindle cells are immunoreactive with CD34 and platelet derived growth factor receptor alpha (PDGFRA), but typically negative for CD21, CD23 and EBER [10]. Inflammatory myofibroblastic tumor (IMT), another mesenchymal tumor with intermediate malignancy, shares similarities with EBV+ iFDCS in histomorphology to a certain extent. More than half of cases manifest ALK translocation, resulting in constitutive tyrosine kinase activation [23]. IMT is composed of bland to mildly atypical, spindled myofibroblasts with a rich inflammatory infiltrate of plasma cells, lymphocytes and eosinophils. Immunophenotypically, IMT cells are positive for smooth muscle actin (SMA) and vimentin, but negative for FDC markers or EBER [10].

Notably, we found that PET/CT showed a nodule in the left thyroid of the patient with increased FDG metabolism, indicating a malignant disease or just a benign change. There have been reports on extranodal FDCS occurring in the thyroid with an inflammatory background of Hashimoto’s thyroiditis [24]. However, the patient’s thyroid function was basically normal, with only a mild elevation of thyroglobulin antibody (4.63 IU/mL, reference range 0–4.11). Although the subsequent thyroid FNA showed a benign lesion, it is necessary to closely follow up the thyroid condition of this patient, given the risk of a missed diagnosis on FNA.

In the process of reviewing the patient’s family history, we learned that her son had been diagnosed with another rare hematologic neoplasm called MLN-EO with FIP1L1-PDGFRA rearrangement manifesting as T-LBL for over a year. Given this, it is worthwhile to explore the genetics in common that are involved in the neoplasms of the mother and son. We intend to clarify the genetic findings in our future studies.

## 4. Conclusions

In conclusion, we have reported a new case of EBV+ iFDCS presenting as a colonic polyp with increased lgG4-positive plasma cell infiltration. Whether lgG4 mediates the physiopathologic mechanism of EBV+ iFDCS needs further investigation. EBV+ iFDCS is an extremely infrequent tumor with distinct histopathological features and immunophenotype. It is of great importance to correctly recognize and diagnose it to avoid inappropriate treatment.

## Figures and Tables

**Figure 1 medicina-59-01341-f001:**
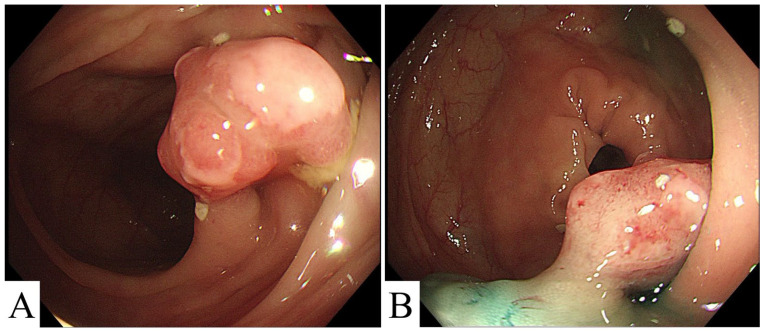
Endoscopy revealed an 18 mm polyp in the transverse colonic lumen. (**A**) The endoscopic morphology of the polyp. (**B**) The lifting sign of the polyp was positive.

**Figure 2 medicina-59-01341-f002:**
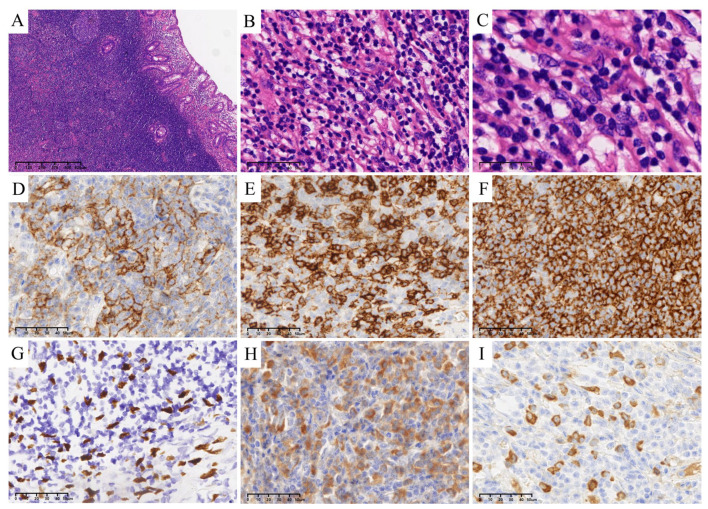
Microscopic examination of the lesion. (**A**–**C**) Hematoxylin and eosin (**H**,**E**): atypical ovoid to spindle tumor cells with vesicular chromatin, clear nuclei but indistinct cell borders scattered in the background of predominantly lymphoplasmacytic infiltration. (**D**–**F**) Immunohistochemistry showed that the tumor cells expressed CD21 (**D**), CD23 (**E**) and CD35 (**F**). (**G**) EBV-encoded mRNA (EBER) ISH showed positive signals on ovoid to spindle tumor cells (the brown color showed the EBV-positive tumor cells). (**H**,**I**) Markedly increased quantities of lgG+ (**H**) and IgG4+ (**I**), plasma cells (>100/HPF) in the scope of the lesion, accompanied by an elevated IgG4: IgG ratio (40~50%).

**Figure 3 medicina-59-01341-f003:**
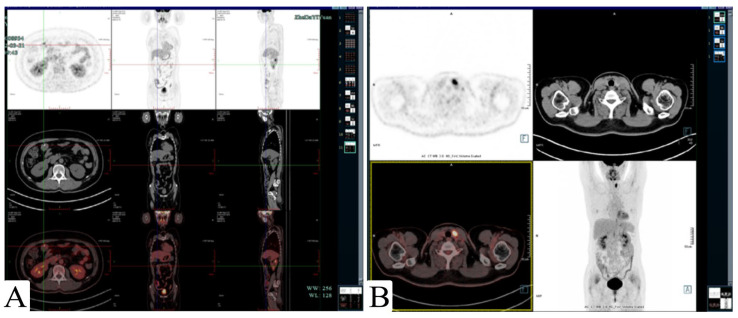
Positron emission tomography/computed tomography (PET/CT): (**A**) a titanium clip is seen in the operative area, and there is no obvious thickening or abnormal radiation uptake in the intestinal wall. (**B**) An approximately 8 × 7 mm slightly low-density nodule in the left thyroid with increased FDG metabolism; the maximum value of the SUV is approximately 14.2.

**Table 1 medicina-59-01341-t001:** Clinical and pathological characteristics of EBV+ inflammatory FDC sarcoma presenting as a colon polyp.

Age/Gender	Clinical Presentation	Colonoscopy Findings	Immunohistochemistry (FDC Markers)	EBER-ISH	lgg4+Plasma Cells and lgg4: lgg Ratio	Treatment	Follow-Up	Source
**78/Female**	Abdominal discomfort, bloody stool; mild anemia (Hb 10 g/dL)	3 cm pedunculated polyp in the transverse colon	CD21+, CD23+, CD35+, D2-40+	Positive	Not mentioned	Polypectomy	Disease free after 5 months	Pan et al., 2014 [9]
**46/Female**	Symptomatic anemia (Hb 6.4 g/dL, dizziness, breathlessness, lethargy); colo-colic intussusception	4 cm broadly pedunculated polyp at hepatic flexure	CD21+, CD23+, CD35+, D2-40+, clusterin+	Positive	Increased (240/HPF); 70%	Biopsy, followed by right hemicolectomy	Disease free after 5 months	Goh et al., 2020 [10]
**54/Male**	Positive fecal occult blood test	2.4 cm pedunculated polyp at the splenic flexure of the colon	CD21+, CD23+, D2-40+, fascin+	Positive	Not mentioned	Endoscopic polypectomy	“Follow-up uneventful”	Chen et al., 2020 [11]
**68/Male**	Positive fecal occult blood test	2.0 cm polyp in the transverse colon	CD21+, CD23+, D2-40+, fascin+	Positive	Not mentioned	Endoscopic polypectomy	“Follow-up uneventful”	Chen et al., 2020 [11]
**53/Male**	Chest and back pain	2 × 2 cm pedunculated polyp 400 mm from the anal verge	CD21+, CD23+, CD35+, D2-40+	Positive	Not mentioned	Endoscopic polypectomy	Disease free after 11 months	Ke et al., 2020 [5]
**48/Male**	Left lower quadrant pain, irregular bowel movement	7 × 4 cm cauliflower-like pedunculated polypoid mass in the transverse colon	CD21+, CD23+, CD35+, D2-40+	Positive	Not mentioned	Right hemicolectomy	Disease free after 7 months	Ke et al., 2020 [5]
**56/Female**	Positive fecal occult blood test	3.2 cm pedunculated polyp in the transverse colon	CD21+, CD23+, CD35+, D2-40+	Positive	Few cells	Surgical resection	Disease free after 14 months	Zhao et al., 2021 [12]
**53/Male**	Abdominal pain and blood in stool	3.5 × 3 × 3 cm pedunculated polyp in the descending colon	CD21+, CD23+, CD35+, D2-40+/−	Positive	Not mentioned	Left hemicolectomy	Disease free after 18 months	Jiang et al., 2021 [8]
**77/Female**	Asymptomatic	2.5 × 3 cm pedunculated polyp in the descending colon	CD21+, CD35+, D2-40+	Positive	Not mentioned	Endoscopic polypectomy	Disease free after 14 months	Jiang et al., 2021 [8]
**59/Female**	Rectal bleeding and anemia	2.5 × 2 × 1.6 cm pedunculated polyp in the proximal descending colon	CD21+, CD23+, CD35+, D2-40+	Positive	Not mentioned	Endoscopic polypectomy	Disease free after 84 months	Jiang et al., 2021 [8]
**57/Male**	Asymptomatic	0.8 × 0.7 × 0.6 cm pedunculated polyp in the sigmoid colon	CD21+, CD35+, D2-40+	Positive	Not mentioned	Endoscopic polypectomy; immunosuppressive therapy for paraneoplastic pemphigus	Died at 7 months from refractory paraneoplastic pemphigus	Jiang et al., 2021 [8]
**64/Female**	Rectal bleeding	2.4 × 1.3 × 1.2 cm pedunculated polyp in the distal ascending colon	CD21+/−, D2-40+	Positive	Not mentioned	Endoscopic polypectomy	Disease free after 122 months	Jiang et al., 2021 [8]
**52/Female**	Asymptomatic	1.8 cm polyp in the transverse colon	CD21+, CD23+, CD35+	Positive	Increased (>100/HPF); 40~50%	Endoscopic polypectomy	Disease free after 15 months	Current case

## Data Availability

Data sharing is not applicable to this article as no datasets were generated or analyzed during the current study.

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
