# Peer review of "Epstein-Barr Virus-Positive Inflammatory Follicular Dendritic Cell Sarcoma Presenting as a Colonic Polyp: Report of a Case with a Literature Review"

_medicina, 2023, doi:10.3390/medicina59071341_

Round 1
Reviewer 1 Report
Medicina
Reviewer’s comments_1:
In this case report, the authors presented the results of a very interesting
case of a 52-year-old woman with a special family history undergoing routine physical exam. It has been well written and discussed. Although, the study design is strong and impressive, but the data presentation could be improved within the scope.
Major comments:
In Fig.2(D-I), higher resolution IHC image (like H&E image presented here in the same Fig.) will improve the quality of the presentation.
In Fig.2(G), different chromogen (i.e., red) should have been used to distinguish IHC chromogen.
Author Response
Dear reviewer:
Please see the attachment.

Reviewer 2 Report
Dear Authors,
I have read with great interest the paper titled: Epstein‒Barr Virus-Positive Inflammatory Follicular Dendritic Cell Sarcoma Presenting as a Colonic
Polyp: Report of a Case with Literature Review
The paper is about a rare neoplasm called Follicular dendritic cell sarcoma (FDCS) that derives from the abnormal proliferation and differentiation of follicular dendritic cells (FDCs). FDCs are mesenchymal-derived cells located in B follicles that serve as antigen-presenting cells and play a major role in the induction and maintenance of the humoral immune response. The paper focuses on a specific type of FDCS called Epstein-Barr virus-positive inflammatory FDCS (EBV+ iFDCS), which used to be classified as the inflammatory pseudotumour (IPT)-like variant type. The paper aims to identify the potential pathogenesis sites of EBV+ iFDCS in extra-hepatosplenic regions for correct and timely diagnosis. The authors also report a case of EBV+ iFDCS presenting as a colon polyp and summarize the clinicopathological features of this tumor type after a thorough review of the literature.
I found interesting the case study and the conclusion + the further investigation needed to go deeper into the topic.
best,
The English is ok according to me.
Author Response

(The authors gave the same response as above.)

Reviewer 3 Report
Comments to Author
1. Please mention the incidence of Epstein‒Barr virus (EBV)-positive in male and female?
2. Please include some points about what are the sources of EBV+?
3. What is the reason that EBV positive in predominant in female then male?
4. If any other drugs are there to suppress or eliminate EBV infection in human, if available please mention?
5. Figure-2, include the scale bar?
Author Response

(The authors gave the same response as above.)

Reviewer 4 Report
A good case report of rare type of polyp.
Minor points;
1) Did endoscopist perform precise observation of the polyp, such as NICE (NBI International Colorectal Endoscopic) Classification ?
2) How the endoscopist decided that the polyp is suitable for endoscopic resection? Was the preoperative diagnosis mucosal cancer or benign polyp?
Author Response

(The authors gave the same response as above.)
